# Genome-Wide Identification, Characterization, and Expression Analysis of Four Subgroup Members of the GH13 Family in Wheat (*Triticum aestivum* L.)

**DOI:** 10.3390/ijms25063399

**Published:** 2024-03-17

**Authors:** Yue Yin, Dongjie Cui, Hao Sun, Panfeng Guan, Hanfeng Zhang, Qing Chi, Zhen Jiao

**Affiliations:** 1Henan Key Laboratory of Ion-Beam Green Agriculture Bioengineering, School of Physics and Microelectronics, Zhengzhou University, Zhengzhou 450052, China; gandalfyy@126.com; 2State Key Laboratory of Cotton Biology, School of Agricultural Sciences, Zhengzhou University, Zhengzhou 450052, China; cuidongjie0323@zzu.edu.cn (D.C.); sunhau@zzu.edu.cn (H.S.); guanpanfeng@zzu.edu.cn (P.G.); hfzhang@zzu.edu.cn (H.Z.); 3Sanya Institute, Zhengzhou University, Zhengzhou 450001, China

**Keywords:** wheat, GH13 family, phylogenetic analysis, subcellular localization, expression analysis, germination

## Abstract

The glycoside hydrolase 13 (GH13) family is crucial for catalyzing α-glucoside linkages, and plays a key role in plant growth, development, and stress responses. Despite its significance, its role in plants remains understudied. This study targeted four GH13 subgroups in wheat, identifying 66 GH13 members from the latest wheat database (IWGSC RefSeq v2.1), including 36 α-amylase (AMY) members, 18 1,4-α-glucan-branching enzyme (SBE) members, 9 isoamylase (ISA) members, and 3 pullulanase (PU) members. Chromosomal distribution reveals a concentration of wheat group 7 chromosomes. Phylogenetic analysis underscores significant evolutionary distance variations among the subgroups, with distinct molecular structures. Replication events shaped subgroup evolution, particularly in regard to AMY members. Subcellular localization indicates AMY member predominance in extracellular and chloroplast regions, while others localize solely in chloroplasts, confirmed by the heterologous expression of *TaSEB16* and *TaAMY1* in tobacco. Moreover, 3D structural analysis shows the consistency of GH13 across species. Promoter cis-acting elements are suggested to be involved in growth, stress tolerance, and starch metabolism signaling. The RNA-seq data revealed *TaGH13* expression changes under drought and submergence stress, and significant expression variation was observed between strong and weak gluten varieties during seed germination using quantitative real-time PCR (qRT-PCR), correlating with seed starch content. These findings demonstrate the pivotal role of GH13 family gene expression in wheat germination, concerning variety preference and environmental stress. Overall, this study advances the understanding of wheat GH13 subgroups, laying the groundwork for further functional studies.

## 1. Introduction

Starch is an important energy-storing carbohydrate, widely found in the seeds, tubers, and roots of various plants and other organs, and has served as a staple crop for humans for over 10,000 years. The cultivation of starchy crops, such as *Triticum aestivum* (wheat), *Zea mays* (corn), *Oryza sativa* (rice), and *Solanum tuberosum* (potato), provided the basis for the development of human civilization. Among them, wheat, which is abundant in starch, was domesticated by humans in the Fertile Crescent, about 12,000 years ago, and became one of the most important staple foods for humans [1]. Starch is composed of straight-chain starch and branched-chain starch; their structures are somewhat different. Amylose is a linear structure consisting of glucose residues linked by α-1,4-glycosidic bonds with a few α-1,6 branches, while the main structure of amylopectin is the same as that of straight-chain starch, except that the chain is shorter and contains a large number of α-1,6 branches [2].

With the continuous deepening of research on starch metabolism in plants, the pathways of starch synthesis and degradation have become clearer, and the key enzymes involved have also been categorized by researchers. Glycoside hydrolases (GHs, EC 3.2.1.x and EC 2.4.1.x) are key enzymes in carbohydrate metabolism that are found in three major kingdoms of life (archaebacteria, eubacteria, and eukaryotes) [3]. Glycosidases and transglycosidases are two major classes of enzymes involved in the synthesis and breakdown of starch and most of these enzymes belong to the GH13 family. It has been noted that GH13 family members mainly hydrolyze α-1,4 glucosidic and α-1,6 glucosidic bonds in starch, with some other members functionally involved in glycosyl transfer [4,5,6]. The GH13 family represents the largest family of GHs and belongs to the clan GH-H, which is believed to share a common ancestor and catalytic machinery with the GH70 and GH77 families [7].

Early studies suggested that the GH13 family was included in the α-amylase family and, as research progressed [8,9,10], more enzymes with other catalytic activities were incorporated into the GH13 family. However, studies of the GH13 family in plants have focused on three classes of α-amylase (AMY) members, starch branching enzymes and de-branching enzymes, with the branching enzymes mainly comprising the 1,4-α-glucan -branching enzyme (SBE), and the de-branching enzymes mainly comprising isoamylase (ISA) and pullulanase (PU) enzymes. Structurally, GH13 enzymes are characterized by a conserved structural core, composed of three domains, often designated as domains A, B, and C. Domain A folds as a (β/α) 8-barrel, domain B is a loop of variable length inserted between strand β3 and helix α3 of the (β/α) 8-barrel. Domain C is a C-terminal extension, characterized by a Greek key structure. Between domains A and B is a catalytically active center, containing three active residues [11,12,13,14,15]. The hydrolase and transferase enzymes that make up the GH13 family both have multiple structural domain proteins, but they all share the same core structural domains. In some studies, researchers have noted that the GH13 family has the most basic catalytic triad plus an arginine residue that is totally conserved [16], which explains why the GH13 family is large and contains numerous functions.

In grains, the energy for seed germination is mainly provided by starch stored in the endosperm, whose breakdown produces the monosaccharides that provide energy for seed germination and seedling growth. Starch synthesis involves numerous transglycosidases that contribute to starch side-chain formation. The expression of GH13 family members during seed germination significantly affects seedling development. In germinating grain seeds, isoforms of AMY are secreted from the living aleurone cells and play a key role in the degradation of storage starch in the non-living starchy endosperm, this process fuels early seedling establishment [17]. Moreover, previous studies showed that seedlings of rice varieties (cvs: Dongdao-4 and Jigeng-88) with higher AMY activity developed more rapidly at low temperatures [18]. In *Arabidopsis thaliana*, the functional limitation of AMY promotes the accumulation of starch granules [19]. For SBE, the loss of the SBE1 function in *Arabidopsis thaliana* results in abnormal seed embryos and the failure of cotyledons to develop normally [20]. Maize (*Zea mays*) SBEI loss of function alters the seed starch structure, leading to reduced starch digestibility and slow coleoptile growth during seed germination [21]. A piece of previous research examined the expression levels of PU during the developmental and germination stages of rice (cvs: 9311 and Wuxiangjing 9) seeds and found that PU expression was at a high level during both time periods [22].

In addition to germination, GH13 is instrumental in the degradation of starch in plants throughout the growth cycle, participating in diverse physiological processes. It has been shown that maize isoamylase SU1 (an isozyme of ISA) knockout mutations lead to an abnormal increase in starch side chains in the kernel [23,24]; whereas, SBE1 and SBE2 knockout mutations affect the synthesis of starch side chains [25]. SBE2 mutants in barley (*Hordeum vulgare*), wheat, and rice (cvs: IR64 and Nipponbare) cause changes in the starch structure in the seed grain [26]. It has been suggested that heat stress in rice (cvs: Nikomaru, Kumasannochikara, Genkitsukushi, and Sagabiyori) leads to elevated *AMY* expression, which responds to the damage caused by high temperatures to the embryo by increasing the content of soluble sugars in the seed [27,28]. Indeed, soluble sugars are widely involved in the plant’s response to abiotic stresses, and starch degradation is an important source of soluble sugars, which explains the decrease in the activity of most starch synthases and the increase in catabolic enzyme activity during abiotic stresses [29].

In the current study, 66 GH13 family members in the whole genome of wheat were identified and divided into four subgroup members, namely TaISA, TaSBE, TaAMY, and TaPU. In addition, GH13 family members’ gene structure, conserved structural domains, and cis-acting elements were predicted; we also analyzed the gene expression pattern in different tissues, at different developmental stages, and under two abiotic stresses, and experimentally explored the differences in the expression levels among strong and weak gluten varieties. This study organizes the expression patterns of GH13 family members in wheat, expands the research base on the GH13 family in plants, and promotes the understanding of genes related to starch metabolism in wheat.

## 2. Results

### 2.1. Identification and Phylogenetic Analysis of GH13 Family Members in Wheat Genome

A total of 67 *TaGH13* candidate genes were identified, using known GH13 members from the *Arabidopsis thaliana* and rice genomes as retrieval targets. After NCBI Conserved Domain Database (CDD) detection, sequences containing structural domains ‘AmyAc_plant_IsoA (CDD Accession: cd11346), E_set_GDE_Isoamylase_N (CDD Accession: cd02856), AmyAc_Glg_debranch (CDD Accession: cd11326), PLN02447, PLN02960, PLN02361, PLN02784, PLN00196 and PLN02887′ were eventually identified (Table 1). Based on the phylogenetic analysis of known members of rice and *Arabidopsis thaliana* gene families and their domain composition, the GH13 family was classified into four subgroups, AMY, SBE, ISA and PU (Figure 1). More specifically, 36 members belonged to the AMY subgroup, 18 members belonged to the SBE subgroup, 9 members belonged to the ISA subgroup, and 3 members belonged to the PU subgroup. Then, we named them *TaISA1-9*, *TaSBE1-18*, *TaAMY1-36*, and *TaPU1-3*, according to their respective chromosomal location and arrangement (Table 1 and Figure 1). The details of the *TaGH13* gene family, including gene ID, amino acid length, molecular weight (MW), isoelectric point (pI), instability index, aliphatic index, and grand average hydropathicity (GRAVY), are provided in Table 1. The number of amino acids of TaGH13 members ranged from 361 (TaSBE7) to 963 (TaAMY19, 36), and the MW ranged from 40,278.79 Da (TaSBE7) to 106,019.4 Da (TaAMY19). The pI value went from 5.31 (TaAMY36) to 8.63 (TaSBE14), most of which were less than 8, suggesting that most TaGH13s were acidic proteins. TaISA8, TaSBE2, TaISA6, TaISA2, TaISA1, TaISA3, TaAMY28, and TaAMY17 have an instability index greater than 40, which proves that these proteins are unstable (instability index > 40 is considered an unstable protein). The aliphatic index and GRAVY value indicated that all of the TaGH13s were hydrophilic proteins, with an aliphatic index < 100 and a GRAVY < 0.

### 2.2. Chromosomal Locations, Synteny Analysis, and Duplication Events of TaGH13 Family

Since wheat is a hexaploid plant containing three subgenomes (A, B, and D), each wheat gene can have orthologues on three homologous chromosomes. Based on the IWGSC database (https://www.wheatgenome.org/, accessed on 18 February 2023), the physical locations of the *TaGH13* genes on the corresponding chromosomes are depicted in Figure 2. Evidently, localization analysis of *TaGH13* chromosomes showed that the *TaGH13* genes were unevenly distributed on 21 chromosomes, including 20 *TaGH13* genes in A subgenome, 25 in B subgenome, and 21 in D subgenome. In particular, homoeologous group 7 chromosomes with 27 *TaGH13* members had the highest density, two of them were closely arranged at the lower part of the chromosomes, but chromosome groups 1, 3, and 4 only contained three *TaGH13* genes, and no *TaGH13* genes were identified on chromosome 4A.

In order to explore the possible mechanism of *TaGH13* expansion, we studied the gene replication events of wheat itself; the red lines can be seen to demonstrate the collinearity of *TaGH13* within the wheat genome (Figure 2). A total of 62 segment repeats and four single copy sequences were identified, indicating that segment repeats played a key role in the expansion of GH13 in the wheat genome. In addition, we found that most *TaGH13* genes had corresponding homologous copies on the A, B, and D chromosomes, implying that the replication events of most genes occur between subgenomes, but there are still a few instances of gene replication within subgenomes (*TaAMY30*, *TaAMY31*, *TaAMY32*, etc.). In addition, non-synonymous (Ka) and synonymous (Ks) substitution of each duplicated *TaGH13* gene was calculated using PAMLX (version 1.3.142). Surprisingly, the calculated Ka/Ks of all duplicated *TaGH13* gene pairs were far less than one (Appendix A), suggesting that the TaGH13 family have probably suffered from strong purification selection in the course of evolution. On the other hand, the distribution of the average time of differentiation of *TaGH13* members is wide, occurring from 0.99 to 45 million years ago (MYA), suggesting that the duplication events occurred before the hybridization events of the A, B, and D subgenomes.

### 2.3. Gene Structure, Motif, and Domain Composition of the TaGH13 Family

To gain insight into the function of the *TaGH13* family in wheat, we analyzed the phylogenetic relationship, gene structure, and conserved motifs (Figure 3). From the phylogenetic tree, based on multiple alignments of full-length protein sequences, we found that all the homologous genes, assigned the same color, were generally more easily clustered into one group than the others (Figure 3A).

Based on the high-quality wheat genome assembly results, we obtained more accurate information about the gene structure. As can be seen in Figure 3B, the gene structures are quite diverse among the *GH13* members with different intron numbers, except for the homologous genes from different subgenomes. The gene constructions became more and more complicated for increasing exons and introns. *TaISA1* to *TaISA3* are very tight, with almost no intron sequences; *TaISA4* to *TaISA9* show an abundance (more than ten) of exons. For *TaSBE*, all members are structurally complex, with half of the introns and exons interspersed with each other, more than 15 times, and a long intron region and, the other half, having up to 10 exons only and a large, segmented exon region. *TaAMY* demonstrates a remarkable level of conservatism, distinctively. All the members, except *TaAMY1-6*, share a striking structural similarity, characterized by an overall compact structure, with minimal intron regions and a significant exon region. *TaPU* is more loosely structured and consists of a large number of exons and introns (more than 25).

To characterize the conserved motifs in the TaGH13 family, the amino acid sequences of the 66 TaGH13 proteins were submitted to the MEME website and the results plotted using TBtools [30,31]. Ten conserved motifs were identified in the proteins, and the distribution of these motifs in TaGH13 is shown in Figure 3C. The motif logos are shown in Appendix A, and it can be seen that the motif distribution patterns and gene structure of TaGH13 members in the same subgroups were similar, but their functions may be diverse. All members of the TaGH13 family contained motif 4, except TaAMY7, and some motifs exist only in specific subgroups, such as motif 8 in the SBE subgroup, and motif 1, 6, and 7, in the AMY subgroup.

The NCBI CDD database (https://www.ncbi.nlm.nih.gov/Structure/cdd/wrpsb.cgi, accessed on 25 July 2023) and the InterPro database (https://www.ebi.ac.uk/interpro/, accessed on 25 July 2023) were used to detect the conserved domains of TaGH13 members. Through the results provided by the NCBI Batch CD-Search tools in the CDD database, it can be seen that nine domains, namely ‘AmyAc_plant_IsoA (CDD accession: cd11346), E_set_GDE_Isoamylase_N (CDD accession: cd02856), AmyAc_ Glg_debranch (CDD accession: cd11326), PLN02447, PLN02960, PLN02361, PLN02784, PLN00196, and PLN02887 (CDD accession: cd11341)’ were detected (Figure 3D). These domains can clearly separate the functional regions of TaGH13 members, for e.g., cd11346 is the characteristic region of ISA, and PLN02447 is the characteristic region of SBE; moreover, through the analysis using the InterPro database, we found that all TaGH13 members contain a ‘Glycosyl hydrolase, family 13, catalytic domain (IPR004193)’, proving that all the members were correctly identified as GH13 family members.

### 2.4. Prediction of Cis-Acting Elements and Transcription Factor (TF) Binding Sites in TaGH13 Promoters

In order to understand the potential function of the *TaGH13* gene family, we analyzed the cis-acting elements of their promoters and found 25 cis-acting elements with annotated functions, which were classified into two major categories, inducible cis-acting elements and transcription factor binding site elements (Figure 4). A total of 13 major inducible cis-acting elements were predicted and these cis-acting elements were related to environmental stress, hormone response, and starch metabolism (Figure 4A). More than 300 elements responded to environmental stresses, including 59 low-temperature, 96 drought-inducible, and 177 anoxic-inducible elements, and more than 900 elements responded to the phytohormone, mainly involving 65 auxin (IAA), 418 jasmonic acid methyl ester (MeJA), 44 salicylic acid (SA), 277 abscisic acid (ABA), and 114 gibberellin (GA) hormones, indicating that the *TaGH13* family play an important role in wheat’s resistance to abiotic stress. Some promoters involve seed or endosperm-specific elements, suggesting that these genes may be associated with wheat grain development. Moreover, the α-amylase promoter element is apparently an AMY-specific element.

Cis-acting elements regulate the precise initiation and efficiency of gene transcription by binding to TFs. As can be seen from Figure 4B, ERF and G2-like binding sites are the most abundant, accounting for almost 60% of the transcription factor binding sites. 

The cis-acting elements mapped to the promoter sequences of the *TaGH13* genes were found to be highly divergent, which further suggested their involvement in different biological processes, hormonal responses, and environmental stresses. It showed that there are always differences in the cis-acting elements between duplicates. All these findings indicated that sub- or neo-functionalization could occur during the duplication process of *GH13* genes in wheat [32]. Interestingly, *TaAMY26-36* all contain an α-amylase signature recognition region near the promoter 3′ region, whereas the other *TaAMY* members do not contain this signature region, indicating that *TaAMY* can also be further differentiated into different subfamilies within *TaAMY*. 

### 2.5. Subcellular Localization of TaGH13 Family Proteins

Subcellular localization prediction using the online tool ProtComp 9.0 (https://www.softberry.com/berry.phtml?topic=protcomppl&group=programs&sub-594group=proloc, accessed on 25 July 2023), demonstrated that TaGH13 may be localized in chloroplasts in all three subgroups, except the AMY subgroup, and the probability of the AMY subgroup being localized in chloroplasts is also close to 40% (Appendix A). To analyze the subcellular localization of these gene products, we cloned and fused the coding sequences of two *TaGH13* genes with the mCherry fluorescent protein, and transiently expressed them in *Nicotiana benthamiana* with the *CaMV35S* promoter. We randomly selected two genes (*TaSBE16*, *TaAMY1*) and performed transient overexpression in tobacco. In order to prevent the low expression of the genes affecting the observation, we designed a recombinant plasmid ‘pC1300::35s::Gene:: mCherry’ based on the pCAMBIA1300 plasmid to enhance the fluorescence signal. As shown in Figure 5, the positions of the chloroplast autofluorescence and mCherry fluorescence overlap, indicating that the two genes were localized in the chloroplasts, which confirms the prediction results.

### 2.6. Three-Dimensional (3D) Structure Analysis of TaGH13 Proteins

The high-level structure of a protein plays a crucial role in determining its properties and catalytic activity. To obtain the 3D structures of TaGH13 members, we used the homology modeling approach in AlphaFold to generate 3D protein models. We randomly selected a TaGH13 member from each subfamily as a representative, to analyze its predicted structure and illustrate the results, as shown in Figure 6 (TaISA5, TaSBE1, TaAMY8, and TaPU1). The results showed that all the proteins contained a distinct (β/α) 8-barrel structure, which is essential for the TaGH13 family (Figure 6). Meanwhile, the structure of each subgroup varied dramatically, which may have contributed to their different catalytic activities. We also compared the 3D structure of TaISA5, TaSBE1, TaAMY8, and TaPU1, with AtISA1, AtSBE2.1, AtAMY1, and AtPU1 of *Arabidopsis thaliana*, respectively. The results showed that the high-level structure of the GH13 family members in wheat and *Arabidopsis thaliana* are very similar despite their taxonomic differences, which reveals conservation in the process of their biological evolution, which may contribute to the investigation on the function of TaGH13 genes.

### 2.7. Analysis of GH13 Expression Patterns Using Online Data

Analysis of RNA-seq data from different organs at different developmental stages revealed that *TaGH13* genes showed different expression patterns in different organs and developmental stages (Appendix A and Figure 7). In general, 66 *TaGH13* genes could be divided into four subgroups. All members were expressed in seeds, especially at the germination stage, and although some of the genes were expressed at low levels, their expression levels were already significant relative to other tissues. Most of the members of *TaISA*, *TaSBE*, and *TaPU* were expressed in varying degrees in the whole plant, whereas most of the members of *TaAMY*, were only expressed at a low level in germinated seeds (Appendix A). In addition, *TaAMY* was mainly expressed in the embryo of seeds, with very little expression in the endosperm, suggesting that amylase at the seed germination stage mainly originates from embryo synthesis, and the endosperm only serves as a storage site for starch (Appendix A). From *TaSBE7* to *TaSBE12*, it is evident that these members exhibit high expression exclusively in the endosperm. This suggests a crucial role of SBE proteins in endosperm function, particularly in starch synthesis and storage. As can be seen from Appendix A, *TaPU* is significantly more expressed in other tissues than in floral organs (spike cell, floret cell, anther cell, et al.). The floral organ develops only during the reproductive growth of the plant, and the floral organ constitutes only a very small part of the plant, so the expression of *TaPU* members in the floral organ is significantly lower than other organs or tissues.

The Genevestigator software (version 9.10.0) was used to demonstrate the expression of GH13 members in the whole development stages of wheat. In Figure 7A, PRJNA529321, PRJNA532455, and PRJNA744310 from the NCBI SRA database were used in our study. In Figure 7B,C, PRJNA604012 from the NCBI SRA database was used in our study. The findings indicate that a significant proportion of GH13 members were actively expressed in seed germination, with certain genes showing notably high expression levels. This underscores the essential function of these genes during the germination stage. As can be seen in Figure 7A, from *TaAMY13* to *17*, these members exhibit low or negligible expression levels during germination. Since these genes evolved from the same ancestral gene, perhaps they only perform functions at stages outside of germination. Other TaAMY subgroup members were completely different, showing extremely high expression during the germination stage, suggesting that TaAMY plays an important role in starch degradation during the wheat germination stage. Since PU is a class of starch-debranching enzymes, *TaPU* expression was also slightly higher at germination than at other developmental stages, which correlates with its function. *TaSBE*, *TaISA*, and *TaPU* expression were slightly elevated at the ‘milk development’ and ‘dough development’ stages, indicating that they are all involved in the process of starch accumulation in seeds.

Plants adjust gene expression in response to abiotic stresses. Through the analysis of transcriptome sequencing data, it was found that all ISA and PU members were expressed in wheat under drought stress, and significant detection of TaSBE2-7, 9, 13-18 expression was also observed (Figure 7B). Only *TaAMY1-6* from the *AMY* subgroup was expressed, while the other genes had no or minimal expression. Submergence stress demonstrated partly the same results as drought stress (Figure 7C). *TaISA*, *TaPU*, and *TaSBE1-6*,*13-18* also manifested higher expression in submergence stress, except that *TaPU* was much more highly expressed in flooded environments than in drought. In addition, the TaAMY subgroup showed a very different expression reaction from drought stress. Almost all genes, except *TaAMY13-17*, had extremely high expression in submergence stress. 

### 2.8. Expression of GH13 Members during Germination in Different Varieties of Wheat

Weak and strong gluten varieties are narrowly defined as wheat varieties, with protein content below 11% and above 35%, respectively, and there are no current reports confirming a correlation between the expression of *GH13* family members and different protein contents during seed germination. Four weak and four strong gluten varieties of wheat were selected for germination experiments; Zhengmai 004, Yangmai 13, Zhengnong 4108, and Chinese Spring are weak gluten varieties; Guohong 6, Womai 9, Zhenmai 168, and Zhengmai 366 are strong gluten varieties. *TaISA2*, *TaISA7*, *TaSBE2*, *TaSBE13*, *TaSBE18*, *TaAMY13*, *TaAMY26*, and *TaAMY19* genes were selected for qRT-PCR analysis. The wheat germination stage is divided into two stages, the first stage in which the seed absorbs water and oxygen (germination stage 1, GS1), and the second stage in which the seed grows a radicle and a germ sheath (germination stage 2, GS2) [33]. We performed qRT-PCR on the seedlings at these two stages, separately (Figure 8).

As can be seen from Figure 8, the expression levels of almost all *TaGH13* genes during the whole germination stage were significantly lower than those of the internal control, and the expression levels of some genes, such as *TaSBE2* and *TaSBE18*, hardly changed in GS1 and GS2. However, the gene expression levels at the two germination stages were significantly different, and the expression levels of the same gene in different varieties were also significantly different. The results for the GS1 stage showed that the expression of *TaAMY19* and *TaAMY26* were lower than the internal control, but significantly higher than all the other genes. *TaISA2* had a significantly higher expression trend in weak gluten varieties than in strong gluten varieties. Except for the above three genes, the expression of the other genes was almost zero, indicating that these genes had not been activated at this stage (Figure 8A). The situation was different at the GS2 stage, where the expression trend of all genes showed up-regulation, which was more pronounced in the weak gluten varieties. Compared with the GS1 stage, the expression of all the genes increased in most of the weak gluten varieties, but the change in expression was not significant in Zhengnong 4018. Meanwhile, the expression of all the genes, except the *TaAMY* gene, remained low in the strong gluten varieties. In addition, it can be seen that the expression of the *TaAMY* gene increased dramatically in the GS2 stage, and the gene expression even exceeded the internal control in Yangmai 13, Chinese Spring, and Zhenmai 168 (Figure 8B).

Overall, these results showed that *GH13* family expression is more active in the germination stage of weak gluten varieties, and also confirms that the *TaAMY* gene is strongly expressed during germination. Some *TaISA* members are also involved in vital activities during the germination stage, and TaSBE is hardly activated during the germination stage.

## 3. Discussion

The glycoside hydrolase family 13, one of the largest families of glycoside hydrolases, is widely found in a variety of prokaryotic and eukaryotic organisms [34]. However, as a class of enzymes that are extremely important in industrial applications, no member of the GH13 family has been systematically studied in wheat. The identification of the *TaGH13* family at the genomic level, using bioinformatic tools, helps improve our understanding of the regulatory function of *GH13* in plant growth and development. For this reason, we utilized existing model species (*Arabidopsis thaliana* and rice) for the identification of *GH13* family members in wheat. Since there is no previous classification of *GH13* family genes to subfamilies in plants, we refer to the clustering of *TaGH13* with existing members of rice and *Arabidopsis thaliana* for classification, mainly focusing on *AMY*, *SBE*, *ISA*, and *PU* [35,36,37,38]. Furthermore, we performed basic genome-wide analysis of the *TaGH13* family, including chromosomal location, subcellular localization, synteny analysis, structure analysis, and promoter cis-acting element analysis. In addition, developmental and tissue-specific expression patterns of wheat *GH13* family members and their differential expression during germination in different wheat cultivars were also explored. These results will lay the foundation for further functional analysis of *TaGH13* families. 

Usually, the pI of all proteins in vivo is around six, which is a weakly acidic, neutral protein. Calculation of the pI, in this study, showed that all 66 TaGH13 members had pIs in the range of 5.31–8.63, with 60 members below 7, proving that most of them are weakly acidic, neutral proteins. It has been shown that subcellular localization correlates with protein pI, with proteins having a pI of 6–8 preferentially being localized extracellularly [39,40], whereas proteins within chloroplasts generally have a pI between 4.4–7 [41]. Our predictions show that most members with a pI between 5.31–7 are ISA, SBE, and PU, which is consistent with the conclusion that they are localized in chloroplasts. In addition, most of the AMY members have a pI between 5.31–6.5, but there are still a few TaAMY members with pIs in the range of 6.5–8.47, confirming their ability to participate in extracellular functions. 

In addition, subcellular localization predictions indicate that members other than TaAMY are localized in chloroplasts, with TaAMY demonstrating a very high probability of extracellular localization, along with a small probability of localization in chloroplasts. We randomly selected two genes (*TaSBE16*, *TaAMY1*) and performed transient overexpression in tobacco, and the results showed that these two genes are clearly localized in chloroplasts. Wang et al. [20] found that SBE1 is localized in chloroplasts in *Arabidopsis thaliana*, which has important an role in embryonic development and starch metabolism, and that *SBE1* is highly conserved in higher plants. Kitajima et al. [42] found that rice (cv: Nipponbare) AMY1 is involved in starch degradation in rice organelles and is localized in plastids, which is consistent with our verified localization in chloroplasts. In addition, Seung et al. [43] stated that AtAMY3 is an α-amylase localized in chloroplasts. These studies show that our findings agree with previous reports. In addition, it is necessary to point out that although our experimental results confirmed the location of AMY in chloroplasts, there are studies showing that AMY also functions extracellularly. Doyle [44] found that AtAMY1 is involved in the degradation of starch after cell death and confirmed that the AMY1 protein is secreted.

We identified 66 TaGH13s in wheat that were classified into four subgroups, *TaISA*, *TaSBE*, *TaAMY*, and *TaPU*, by analyzing their structural motif and referring to the classification of the *GH13* family in *Arabidopsis thaliana* and rice. It is known that there are ten *GH13* members in *Arabidopsis thaliana* and 12 in rice, whereas we identified that the number of *GH13* members in wheat is six times higher than that, which may be related to the complex genome of wheat. Studies have shown that gene duplication events have occurred in 70–80% of angiosperms [45,46,47,48,49]. Wheat, an allohexaploid containing three subgenomes, has more than 85% duplicated sequences [50]. Our study found that all *GH13* members were subject to duplication events and multiple members contain over three homologous genes (Figure 3A). The chromosomal localization of the genes suggests that polyploidization, tandem, and segmental duplications may be jointly involved in the formation of the *TaGH13* family (Figure 1). 

Previous studies have shown that the allohexaploid wheat subgenomes, A, B, and D, were originally derived from three diploid species (AA: *Triticum Urartu*, BB: *Aegilops speltoides*, and DD: Aegilops *tauschii*) and underwent three hybridization events [51]. The A and B subgenomes diverged from a common ancestor about 7 MYA and the first hybridization occurred around 5.5 MYA between the A and B subgenomes, leading to the D subgenome through homoploid hybrid speciation [52]. The second hybridization between the A and B subgenomes gave rise to the AABB genome about 0.8 MYA via polyploidization. Wheat originated about 0.4 MYA through allopolyploidization from a third hybridization. By estimating the approximate dates of the segmental duplication pairs of the *TaGH13* genes, we infer that ten paralogous gene pairs in *TaGH13* originated before the first hybridization of the A and B subgenomes (>7.5 MYA). Other paralogous gene pairs occurred after the hybridization event, approximately 0.99–7.36 MYA. Thus, the segmental duplication event was the main driver of *TaGH13* gene evolution during the speciation of allohexaploid wheat, with polyploidization being a complementary means.

The gene structures of *TaISA* and *TaSBE* are more complex and vary considerably among different members. The structure of *TaAMY* is relatively stable and, except for *TaAMY4-6*, all other members are very similar. Things are different in other plants. In rice, the structure of α-amylase varies considerably, with some members containing extraordinarily long intron regions, others with significantly fewer exons than average, and even some with 5′ untranslated region (5′ UTR) lengths exceeding that of the promoter [53]. Yang et al. [54], on the other hand, presented very interesting results in which they categorized the cassava α-amylase family genes into three clusters, each of which is internally structurally conserved. This may indicate that α-amylase conservation in wheat is rare.

The conserved motifs assay showed that TaISA, TaSBE, and TaPU all have the same motif 4 and motif 8, despite their different functions. They differ in that TaISA varies considerably before motif 4, which may or may not be accompanied by other motifs, while TaSBE has a different structure, not only is motif 10 before motif 4, but motif 5 may be after motif 8; TaPU is relatively simple in structure and has motif 3 before motif 4. The similarity in motif structure implies that the three subgroups are functionally related. SBE is a glucanotransferase that generates branch points by cutting an existing α-1,4-linked chain and transferring the cut segment to another linear chain to create a new α-1,6 linkage [55]. Both classes of ISA and PU can hydrolyze α-1,6 branch points, but show different substrate specificities, which probably reflects their different roles in starch metabolism [38]. The motifs 1–7 of all the members of TaAMY are sequentially arranged, showing a very conserved motif structure, which is a reminder that despite the large number of members in the AMY subgroup, it is very conservative in function and structure [38,56].

We found that different subgroups of *TaGH13* have different tissue-specific expressions. In TaISA, most members exhibited stable expression levels throughout the plant. This indicates that *TaISA* genes have an important function throughout the reproductive period. It has been noted that *Arabidopsis thaliana* requires only *ISA1* expression to bring endosperm starch to near normal levels [57], which seems to corroborate the reason for the high expression of the homologous genes, *TaISA4* to *TaISA6*, in the endosperm in our results. SBE is different, with a majority of members stably expressed throughout the plant, which may be related to its involvement in the starch synthesis function in chloroplasts [58]. A number of members (*TaSBE7* to *TaSBE12*) are centrally expressed in the endosperm and exhibit high expression levels, indicating that the members are involved in starch synthesis in the endosperm, which may correspond to the relevant studies in rice and maize [26]. AMY, as an amylase, usually acts as a secreted protein to degrade starch in the endosperm during seed germination and is usually transported to the endosperm to participate in the reaction after being produced by endosperm epithelial cells [59], but it has also been shown that rice (cv: Nipponbare) *AMYI-1* is mainly involved in the degradation of starch in chloroplasts [60]. PU is widespread and stably expressed throughout the plant due to its involvement in the biosynthesis of straight-chain starch, and its synthesis of straight-chain starch is much smaller than that of ISA [61].

The distribution of *TaGH13* expression during development is also well characterized. *TaSBE7-12* are expressed only at later stages of milk development (Figure 7), suggesting that they may be involved only in starch accumulation in the endosperm, which corresponds with their expression only in the endosperm (Appendix A). *TaISA*, *TaPU*, and all *TaSBE*, except *TaSBE7-12*, were stably expressed almost throughout the reproductive period (Figure 7), suggesting that they are allosterically involved in the starch pathway of chloroplast synthesis during wheat growth. *TaAMY* shows an irreplaceable function during seed germination, degrading a large amount of starch in the endosperm to provide energy for seed germination and, at this time, almost all members of *TaAMY* have high expression levels. The exceptions were four genes, *TaAMY13-16*, which were expressed at slightly higher levels at the anthesis and milk development stage, but not expressed at other stages. Considering that these four genes evolved from the same ancestor, it is implied that their functions are not related to starch degradation during seed germination [42,62]. As mentioned earlier, it has been suggested that rice (cv: Nipponbare) *AMYI-1* mainly acts on starch in chloroplasts [60], and perhaps these four genes function in the same way with rice *AMYI-1*.

Several abundant, predicted cis-acting elements are known to mediate plant stress tolerance. For example, *WRKY*, *B3*, and *MYB* [63,64,65] are widely involved in various stress tolerance pathways in plants; C_2_H_2_ is involved in plant tolerance to low temperatures [66] and *AP2/ERF* is mainly involved in multiple abiotic stress resistance pathways [67]. The promoter regions of *TaGH13* members contain multiple transcription factor binding sites, so members of the GH13 family are also extensively involved in regulatory pathways in response to abiotic stresses. Cis-acting element analysis showed that the *TaGH13* promoter region has many elements responsive to drought, hypoxic stresses, suggesting that the expression pattern of *TaGH13* genes may be altered when stimulated by stress. It has been shown that rice (cv: Kranti) α-amylase expression decreases when subjected to osmotic stress [68]. Xiao et al. [69] showed that the inhibition of wheat α-amylase activity improves salt and drought tolerance in wheat. Gilding et al. [70] demonstrated that a low-frequency allele type of the pullulanase locus increases digestibility without tradeoffs in the form of negative pleiotropic effects, thus adding value to a crop pre-adapted to drought and heat stress. In addition, soybeans undergoing flooding stress leads to a dramatic increase in *AMY* and *ISA* expression [71], whereas rice (cv: Nampyeonbyeo) expression of *SBE* and *PU* is markedly increased when subjected to flooding stress to generate more soluble sugars to improve plant tolerance [72].

## 4. Materials and Methods

### 4.1. Identification and Characteristics of GH13 Genes in Wheat

To identify the wheat *GH13* genes, 12 *Oryza sativa* (rice) and 10 *Arabidopsis thaliana* experimentally validated members of the *GH13* family were obtained from Uniprot (https://www.uniprot.org/, accessed on 20 March 2023) (Table 2). The genomes and gene annotations of wheat, International Wheat Genome Sequencing Consortium (IWGSC) RefSeq v2.1, were download from Ensembl Plants (http://plants.ensembl.org, accessed on 20 March 2023) [73]. The TaGH13 members were identified using BLASTP. Firstly, the obtained GH13 protein sequences of rice and *Arabidopsis thaliana* were used for the BLASTP operation (E-value ≤ 1 × 10^−5^) [74,75] to retrieve possible TaGH13 members in wheat from the IWGSC RefSeq v2.1 using TBtools [31]. Then, the NCBI Batch CD-Search (https://www.ncbi.nlm.nih.gov/Structure/cdd/wrpsb.cgi, accessed 20 on March 2023) was used to confirm whether candidate TaGH13 members contained certain domains, namely AmyAc_plant_IsoA (CDD accession: cd11346), E_set_GDE_Isoamylase_N (CDD accession: cd02856), AmyAc_Glg_debranch (CDD accession: cd11326), PLN02447, PLN02960, PLN02361, PLN02784, PLN00196, and PLN02887 [76]. Finally, all the obtained sequences were entered into the InterPro database (https://www.ebi.ac.uk/interpro/, accessed on 20 March 2023) to be used to detect the inclusion of the ‘Glycosyl hydrolase, family 13, catalytic domain’(IPR006047) conserved structural domain [77]. The molecular weight (kDa), theoretical pI of the putative peptides, instability index, aliphatic index, and grand average hydropathicity were calculated using ExPASy (http://www.expasy.ch/tools/pi_tool.html, accessed on 20 March 2023) [78].

### 4.2. Chromosomal Location, Synteny Analysis, Phylogenetic Relationships, and Gene Duplication Analysis of TaGH13 Genes

All candidate genes were localized to different chromosomes, according to the chromosomal information and gene IDs reported in IWGSC RefSeq v2.1. The collinear block was identified by *TaGH13* duplication events in MCScanX [79] and the visualization was carried out using TBtools.

Multiple sequence alignments and the phylogenetic relationship analysis of *GH13* gene families, including intra species and inter species, were performed using MEGA X using the maximum likelihood (ML) method, and 1000 bootstrap replicates. Meanwhile, the non-synonymous (Ka) and synonymous (Ks) substitution in paralogous and/or orthologous gene pairs from 3 species were also estimated using the bioinformatics software PAMLX (version 1.3.142) [80]; the approximate divergence time between duplicated gene pairs was calculated by using the formula T = Ks/2r × 10^−6^, assuming a substitution rate (r) of 6.5 × 10^−9^ substitutions/synonymous site/year [51,81].

### 4.3. Sequence Analysis, Motif Analysis, Cis-Acting Elements Analysis, and 3D Structure Analysis of TaGH13 Members

By extracting the information from the annotation file of IWGSC RefSeq v2.1 with the genome file, we obtained information about the position of introns and exons of *TaGH13* within the gene and visualized them using TBtools. The Multiple Expectation Maximization for Motif Elicitation (MEME) online program (http://meme.sdsc.edu/meme/itro.html, accessed in 20 March 2023) was performed to identify conserved motifs of TaGH13 proteins [30]. Due to the excessively long UTR of some genes, we selected 2500 bp genomic DNA sequences upstream of the start codon of *TaGH13* members as the promoter sequences to analyze the cis-acting elements using the database PlantCARE (http://bioinformatics.psb.ugent.be/webtools/plantcare/html, accessed in 21 March 2023) [82]. 

We used Alphafold2 (https://www.alphafold.ebi.ac.uk/, accessed 25 in March 2023) to predict the 3D structure of TaGH13 [83]. AlphaFold takes multiple sequence alignments and the target sequence as input, predicts the distances between residue pairs in the protein, and then generates the 3D structure of the protein.

### 4.4. Subcellular Localization Analysis

The subcellular localization was first predicted using the ‘ProtComp 9.0’ online tool (https://www.softberry.com/berry.phtml?topic=protcomppl&group=programs&subgroup=proloc, accessed 20 in March 2023), by entering the resulting TaGH13 family member protein sequences. Then, based on the sequences above, we cloned the CDS sequences of *TaSBE16* and *TaAMY1* and constructed them into a transient expression vector with a red fluorescent label as the structure ‘PC1300::*TaSBE16/TaAMY1*::mCherry’. Then, two pairs of genes we selected were transformed into *Agrobacterium tumefaciens* (strain: GV3101). The transient expression of subcellular localization of wheat GH13 members in tobacco plants (*Nicotiana benthamiana*) was assessed using the method from Wu et al. [84], and the mCherry protein was observed under a Zeiss LSM880 laser confocal microscope. All primers are listed in Appendix A.

### 4.5. Expression Analysis of TaGH13 Genes Using In Silico Methods and qRT-PCR

The expression of *TaGH13* members in different tissues, developmental stages, drought, as well as submergence stresses, was obtained from the Genevestigator library (https://genevestigator.com, accessed in 20 March 2023), plotted as a heat map and presented in this study. Eight different varieties of wheat, including four weak gluten and four strong gluten varieties, were selected for expression validation using qRT-PCR. The weak gluten varieties were Zhengmai 004, Yangmai 13, Zhengnong 4108, and Chinese Spring; the strong gluten varieties were Guohong 6, Womai 9, Zhenmai 168, and Zhengmai 366. The seeds were sterilized using a hot-water treatment (55 °C water bath 10–15 min) and then spread on 9 cm Petri dishes lined with filter paper at the bottom for germination [85]. The total RNA was extracted from different samples using Trizol (Takara, Kusatsu, Japan). The qRT-PCR was performed using Tubulin (XM_044534775) and GAPDH (XM_044557128) as the internal controls. The whole wheat seedlings were harvested from different germination stages for expression profile analysis. The results were presented as heat maps and histogram. All the expression levels represent the mean ± SD of the data collected from three independent experiments with three replicates; * for *p* < 0.05 and ** for *p* < 0.01. The subsequent qRT-PCR analysis was used with the 2^−ΔCt^ method [86]. The primers are listed in Appendix A. These experiments were performed in Henan Key Laboratory of Ion-beam Green Agriculture Bioengineering at Zhengzhou University on 15 November 2023.

## 5. Conclusions

In this study, we identified 66 members of the *GH13* family in wheat and demonstrated that they can be categorised into four subgroups. Based on the gene structure and conserved domains, members of the same subgroup may have similar functions. We performed phylogenetic, cis-acting element, conserved motif, and intron–exon analysis, as well as chromosomal location, synteny, and promoter region analysis on the *TaGH13* members. Wheat polyploidization is the main reason for the expansion of *TaGH13* gene family members. The expression pattern of *TaGH13* genes in wheat has tissue specificity and diversity in different development stages. We also verified that high starch content varieties of wheat had higher overall *GH13* gene expression during germination. Overall, this study provides a basis for the study of functional differences in the role played by wheat GH13 family members during seed germination and in response to different stresses.

## Figures and Tables

**Figure 1 ijms-25-03399-f001:**
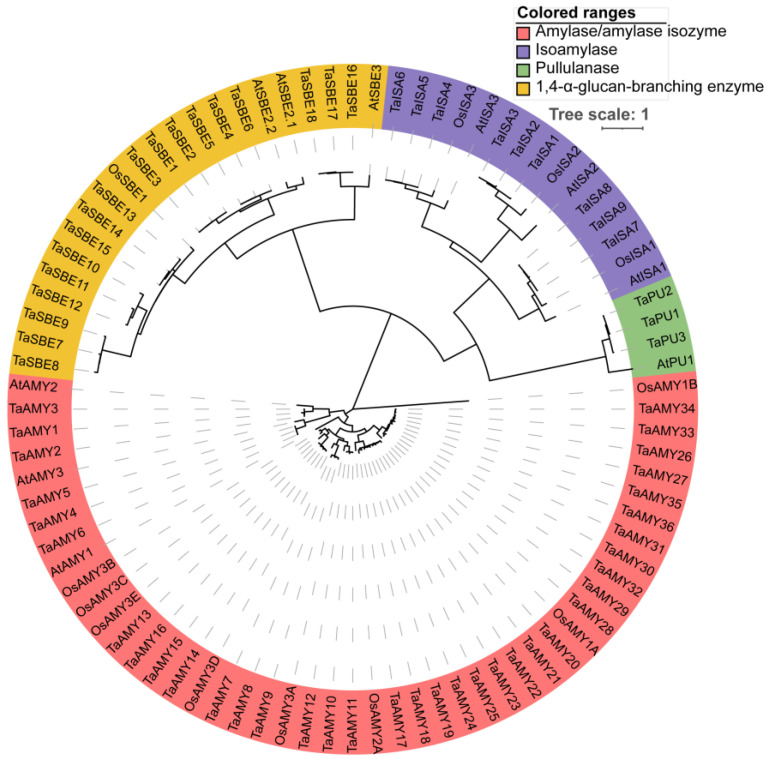
Phylogenetic analysis of protein sequences of the GH13 family in wheat, rice, and *Arabidopsis thaliana*. The phylogenetic tree was inferred to test the maximum likelihood with 1000 bootstraps using MEGA X. Different color backgrounds indicate the clustering of genes in different subgroups, and branch length indicates the difference between different genes; further distance indicates a greater difference between the two genes. The abbreviations in Figure 1: Ta, *Triticum aestivum*; Os, *Oryza sativa*; At, *Arabidopsis thaliana*; AMY, amylase/amylase isozyme; SBE, 1,4-α-glucan-branching enzyme; ISA, isoamylase, and PU, pullulanase.

**Figure 2 ijms-25-03399-f002:**
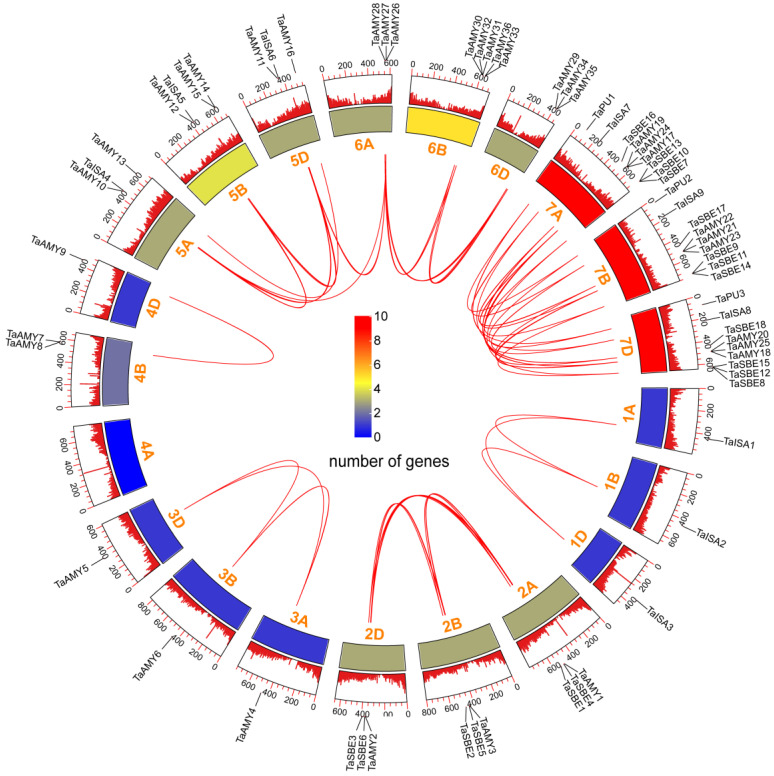
*TaGH13* chromosome localization and collinearity relationships. The inner circle shows the number of *TaGH13* members localized to the current chromosome in the form of a heat map, and the outer circle shows the density of the gene distribution in the wheat genome in the form of a bar chart. The collinearity of *TaGH13* is marked by a red line.

**Figure 3 ijms-25-03399-f003:**
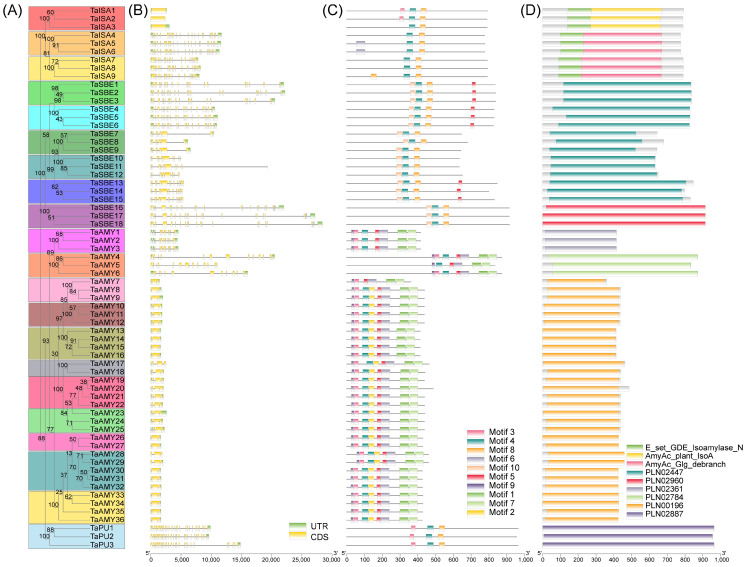
Analysis of phylogenetic relationships, gene structure, conserved motifs, and domains in TaGH13 family members. (**A**) Maximum likelihood tree of TaGH13 members; the same color indicates the homologous genes. (**B**) Gene structure of *TaGH13* members. Exons and introns that belong to the coding sequence (CDS) are represented by yellow boxes and black lines, respectively. The untranslated region (UTR) is represented by green boxes. The ratio of bar and line lengths is consistent with that of exons and introns. (**C**) Conserved motif distribution of TaGH13 members. Each motif is represented by a colored box. (**D**) Domains predicted in TaGH13 proteins. Each domain is represented by a colored box.

**Figure 4 ijms-25-03399-f004:**
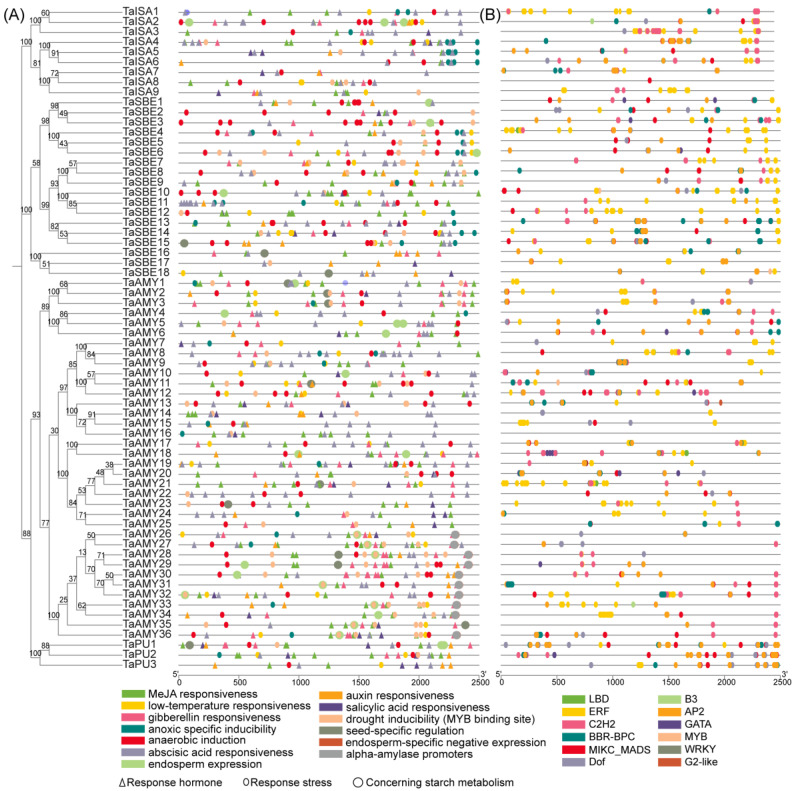
Investigation of inducible cis-acting element and transcription factor binding site elements of *TaGH13* members in wheat. (**A**) Inducible cis-acting element of *TaGH13* members. (**B**) Transcription factor binding site elements of TaGH13 members. Elements in each category are represented by different colors. Δ: response hormone element; ０: response stress element; Ｏ: concerning starch metabolism. A number of elements related to the ‘concerning starch metabolism’ category are too close to each other and overlap at the same point.

**Figure 5 ijms-25-03399-f005:**
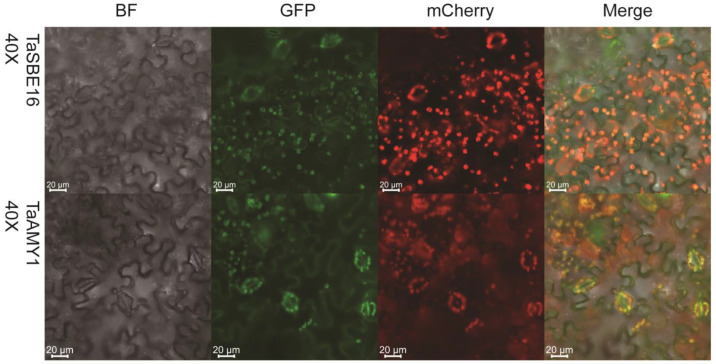
Subcellular localization of TaSBE16 and TaAMY1 in tobacco leaf cells. BF: bright field; mCherry: 580 nm exciting light demonstrated target protein; GFP: 488 nm exciting light demonstrated chloroplast autofluorescence; Merge: overlay of BF, mCherry, and chloroplast image. Scale bar = 20 μm.

**Figure 6 ijms-25-03399-f006:**
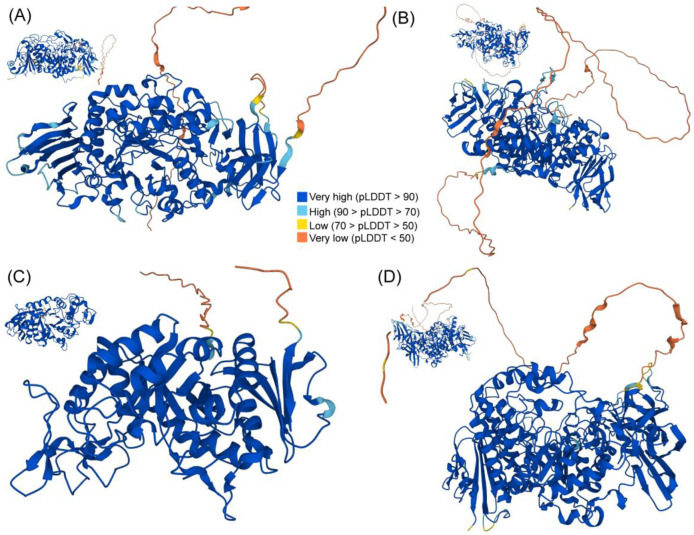
The 3D structure of TaGH13 members. The (β/α) 8-barrel structure can be seen clearly in each figure. Each subfigure represents the protein structure of a homologous protein in *Arabidopsis thaliana*. (**A**) TaISA5; (**B**) TaSBE1; (**C**) TaAMY8; and (**D**) TaPU1. AlphaFold produces a per-residue model confidence score (pLDDT) between 0 and 100. Some regions below 50 pLDDT may be unstructured in isolation.

**Figure 7 ijms-25-03399-f007:**
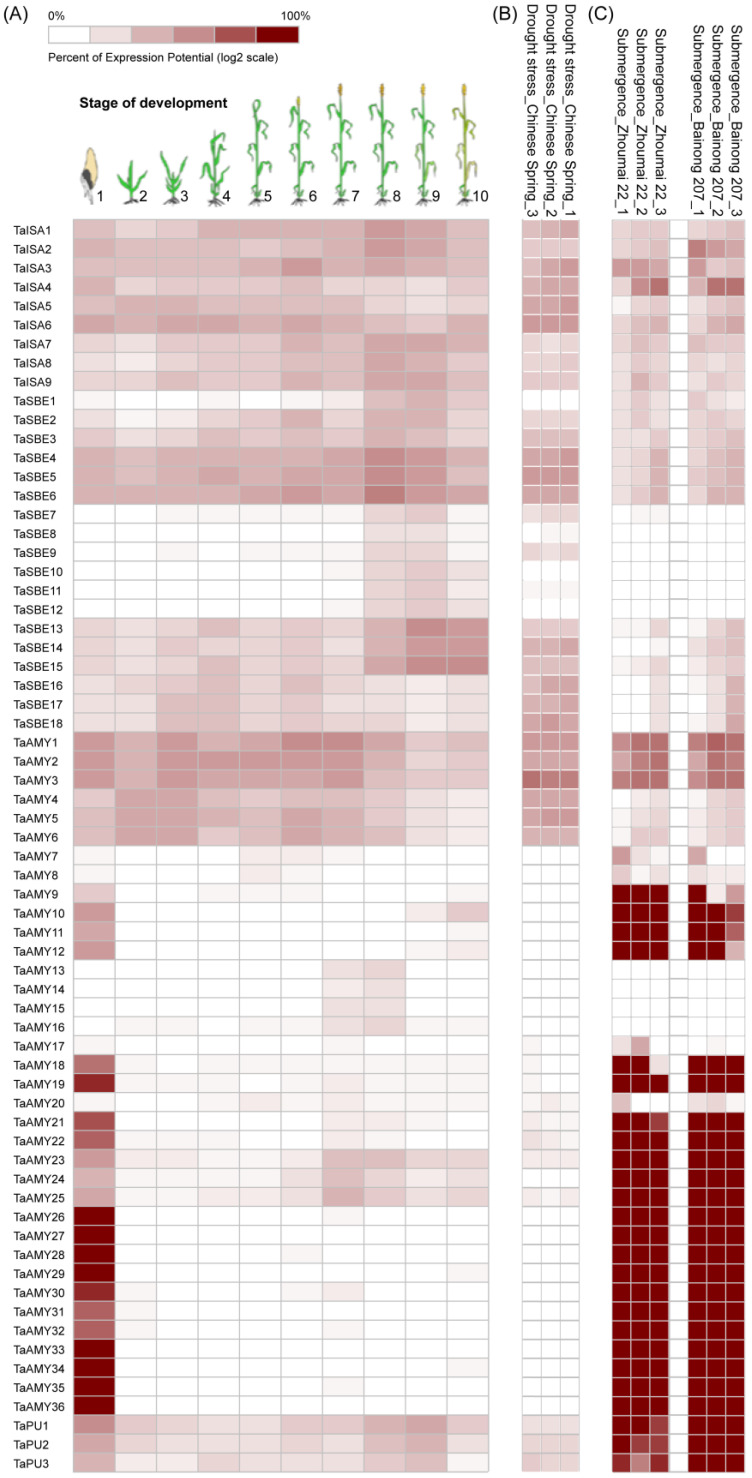
Expression patterns of *TaGH13* members in different tissues at different developmental stages and abiotic stresses. (**A**) Expression pattern of *TaGH13* members at different developmental stages in Chinese Spring. The 10 stages are: ‘1. germination’, ‘2. seedling growth’, ‘3. tillering’, ‘4. stem elongation’, ‘5. booting’, ‘6. inflorescence emergence’, ‘7. anthesis’, ‘8. milk development’, ‘9. dough development’, and ’10. ripening’. (**B**) Expression pattern of *GH13* genes in Chinese Spring seedlings suffering from drought stress. (**C**) Expression pattern of *GH13* genes in two varieties of wheat, Zhoumai 22 and Bainong 207, subjected to flooding stress during the germination stage. The darker color indicates a higher prospect of expression. Data from Genevestigator (version 9.10.0).

**Figure 8 ijms-25-03399-f008:**
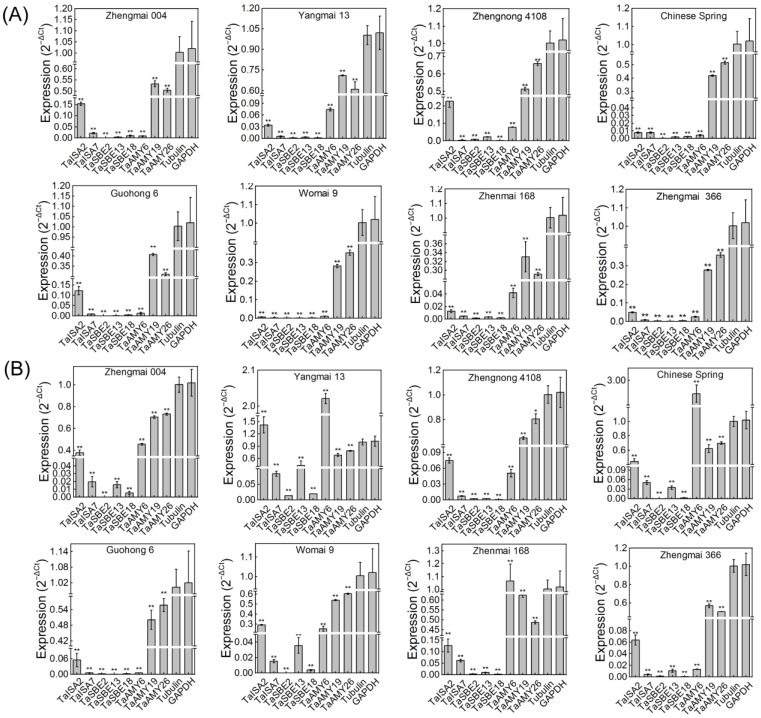
Expression of some *TaGH13* genes during germination of different wheat varieties. Zhengmai 004, Yangmai 13, Zhengnong 4108, and Chinese Spring are weak gluten varieties, and Guohong 6, Womai 9, Zhenmai 168, and Zhengmai 366 are strong gluten varieties. (**A**) Expression of seed imbibition time (GS1) point is shown. (**B**) Expression at the time the seed grows a radicle and a germ sheath (GS2) is shown. Data were calculated as the n-fold difference relative to the internal control (2^−ΔCt^, where ΔCt represents the difference in the threshold cycle between the target and control genes; Tubulin and GAPDH were selected to be the internal control; * for *p* < 0.05 and ** for *p* < 0.01).

**Table 1 ijms-25-03399-t001:** Physical and chemical property of TaGH13s identified in wheat. All gene IDs from Ensembl Plants (http://plants.ensembl.org/index.html, accessed on 3 March 2023).

Name	Gene ID	Number of Amino Acids	Molecular Weight	Theoretical pI	Instability Index	Aliphatic Index	Grand Average Hydropathicity
TaISA1	TraesCS1A02G247100	793	85,711.73	6.31	45.6	86.09	−0.03
TaISA2	TraesCS1B02G257700	788	85,511.58	6.46	47.11	87.74	−0.024
TaISA3	TraesCS1D02G246300	789	85,662.71	6.75	45.42	85.65	−0.056
TaISA4	TraesCS5A02G248700	775	86,074.49	6.1	35.38	74.12	−0.428
TaISA5	TraesCS5B02G246400	776	86,254.8	6.13	35.72	73.53	−0.432
TaISA6	TraesCS5D02G255800	776	86,226.7	6.07	35.56	73.16	−0.439
TaISA7	TraesCS7A02G251400	790	88,718.4	5.72	40.26	69.59	−0.343
TaISA8	TraesCS7D02G249500	791	88,822.5	5.65	39.91	69.53	−0.341
TaISA9	TraesCS7B02G139700	790	88,715.53	5.84	41.48	69.97	−0.333
TaSBE1	TraesCS2A02G310300	834	94,039.27	5.71	39.16	65.32	−0.511
TaSBE2	TraesCS2B02G327300	836	94,367.65	5.56	38.15	66.67	−0.499
TaSBE3	TraesCS2D02G308600	836	94,433.73	5.63	39.65	65.41	−0.511
TaSBE4	TraesCS2A02G293400	828	93,628.97	5.4	38.9	70.53	−0.481
TaSBE5	TraesCS2B02G309500	828	93,662.12	5.46	37.7	71.12	−0.475
TaSBE6	TraesCS2D02G290800	825	93,184.56	5.49	37.3	71.14	−0.46
TaSBE7	TraesCS7A02G549300	645	72,979.6	7.16	37.76	74.7	−0.465
TaSBE8	TraesCS7D02G535600	680	76,951.02	6.61	37.44	76.59	−0.432
TaSBE9	TraesCS7B02G472300	642	72,473.91	6.59	36.62	74.3	−0.474
TaSBE10	TraesCS7A02G549200	635	71,927.42	6.27	34.31	69.09	−0.414
TaSBE11	TraesCS7B02G472400	632	71,376.67	6.27	35.71	70.68	−0.369
TaSBE12	TraesCS7D02G535500	650	73,492.25	6.55	35.56	71.12	−0.35
TaSBE13	TraesCS7A02G549100	846	95,331.37	6.43	34	64.81	−0.501
TaSBE14	TraesCS7B02G472500	798	90,203.54	6.11	31.37	66.74	−0.479
TaSBE15	TraesCS7D02G535400	830	93,577.25	6.27	33.07	64.76	−0.522
TaSBE16	TraesCS7A02G336400	914	105,036.2	6.23	39.21	70.51	−0.547
TaSBE17	TraesCS7B02G248000	914	104,905.1	6.24	38.97	70.62	−0.536
TaSBE18	TraesCS7D02G344000	914	104,889	6.19	39.29	70.73	−0.537
TaAMY1	TraesCS2A02G289800	415	47,254.95	5.44	47.36	70.02	−0.52
TaAMY2	TraesCS2D02G287800	415	47,111.72	5.44	47.59	70.96	−0.504
TaAMY3	TraesCS2B02G306400	415	47,225.89	5.44	48	70.96	−0.518
TaAMY4	TraesCS3A02G248000	871	97,498.62	5.48	38.15	75.36	−0.452
TaAMY5	TraesCS3D02G248000	833	93,182.72	5.62	36.57	74.95	−0.46
TaAMY6	TraesCS3B02G276700	871	97228.37	5.63	38.5	74.35	−0.459
TaAMY7	TraesCS4B02G285600	361	40,278.79	6.22	33.29	85.07	−0.194
TaAMY8	TraesCS4B02G285400	438	48,330.78	6.24	34.77	81.74	−0.184
TaAMY9	TraesCS4D02G284400	437	48,337.83	6.75	36.28	82.17	−0.194
TaAMY10	TraesCS5A02G238100	436	47,610	5.95	29.27	81.67	−0.189
TaAMY11	TraesCS5D02G245000	436	47,656.04	6.27	29.49	80.32	−0.228
TaAMY12	TraesCS5B02G236600	436	47,570.97	6.08	28.92	81.88	−0.204
TaAMY13	TraesCS5A02G464500	413	45,370.5	8.01	26.06	78.64	−0.296
TaAMY14	TraesCS5B02G476000	413	45,458.6	8.63	26.66	79.35	−0.298
TaAMY15	TraesCS5B02G475700	413	45,349.51	8.62	25.91	80.29	−0.283
TaAMY16	TraesCS5D02G477100	413	45,461.68	8.47	26.08	80.07	−0.303
TaAMY17	TraesCS7A02G384000	463	50,860.8	6.14	27.1	80.54	−0.158
TaAMY18	TraesCS7D02G380500	440	48,343.83	5.47	22.65	83.64	−0.137
TaAMY19	TraesCS7A02G383200	437	47,584.79	5.69	20.54	78.65	−0.15
TaAMY20	TraesCS7D02G379700	486	53,511.52	5.66	22.18	75.72	−0.203
TaAMY21	TraesCS7B02G286100	438	47,717.9	5.64	20.54	79.16	−0.152
TaAMY22	TraesCS7B02G286000	438	47,809.05	5.66	20.2	80.25	−0.155
TaAMY23	TraesCS7B02G286700	439	47,989.28	5.48	20.79	80.73	−0.144
TaAMY24	TraesCS7A02G383900	437	47,788.09	5.6	20.08	78.19	−0.161
TaAMY25	TraesCS7D02G380400	437	47,877.09	5.69	20.6	77.76	−0.182
TaAMY26	TraesCS6A02G334200	427	47,378.47	5.83	22.1	81.1	−0.306
TaAMY27	TraesCS6A02G334100	427	47,249.36	6	19.86	81.33	−0.281
TaAMY28	TraesCS6A02G319300	459	50,862.44	5.97	23.11	80.15	−0.323
TaAMY29	TraesCS6D02G298500	459	50,789.34	5.89	22.45	81	−0.302
TaAMY30	TraesCS6B02G349500	425	47,165.24	5.84	20.07	81.27	−0.3
TaAMY31	TraesCS6B02G349800	425	47,209.29	5.84	20.18	81.51	−0.302
TaAMY32	TraesCS6B02G349700	425	47,209.29	5.84	20.07	81.27	−0.307
TaAMY33	TraesCS6B02G364900	427	47,332.49	6.01	22.08	81.36	−0.301
TaAMY34	TraesCS6D02G313300	427	47,346.51	5.92	23.27	81.57	−0.295
TaAMY35	TraesCS6D02G313500	427	47,322.52	6.1	20.52	79.06	−0.294
TaAMY36	TraesCS6B02G364800	425	47,093.21	5.92	20.59	81.27	−0.291
TaPU1	TraesCS7A02G133500	963	106,019.4	5.57	34.03	79.41	−0.275
TaPU2	TraesCS7B02G034600	955	104,967	5.44	32.5	80.69	−0.263
TaPU3	TraesCS7D02G133100	963	105,770.9	5.31	33.16	79.29	−0.266

**Table 2 ijms-25-03399-t002:** List of experimentally validated members of the 12 Oryza sativa and 10 Arabidopsis thaliana GH13 family. All gene IDs from Ensembl Plants.

Gene ID	Name
At4g25000	AtAMY1
At1g76130	AtAMY2
At1g69830	AtAMY3
At2g36390	AtSBE2.1
At5g03650	AtSBE2.2
At3g20440	AtSBE3
At5g04360	AtPU1
At2g39930	AtISA1
At1g03310	AtISA2
At4g09020	AtISA3
Os02g0765600	OsAMY1A
Os01g0357400	OsAMY1B
Os06g0713800	OsAMY2A
Os09g0457400	OsAMY3A
Os09g0457600	OsAMY3B
Os09g0457800	OsAMY3C
Os08g0473900	OsAMY3D
Os08g0473600	OsAMY3E
Os08g0520900	OsISA1
Os05g0393700	OsISA2
Os09g0469400	OsISA3
Os06g0726400	OsSBE1

## Data Availability

Data are contained within the article and Appendix A.

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
