# Peer review of "Genome-Wide Identification, Characterization, and Expression Analysis of Four Subgroup Members of the GH13 Family in Wheat (Triticum aestivum L.)"

_ijms, 2024, doi:10.3390/ijms25063399_

Round 1

Reviewer 1 Report

Comments and Suggestions for Authors

Dear authors,

The manuscript is interesting, but some improvements and the clarification of some ambiguities are necessary to increase the value.

As suggested below:

Line 87: You mentioned that: <In Arabidopsis, functional limitation of AMY promotes the accumulation...>

Sometimes you write the name of the Arabidopsis genus, italicized, sometimes not italicized. You should keep the same single line. as a rule, the name of the genus is not italicized, only the name of the entire species (but which is missing here)

However, what species is it? Here and throughout the manuscript, only the genus Arabidopsis is treated, although there are many species included. As such, please mention which section you are referring to.

Line 117: Sometimes you write the name of the Arabidopsis genus, italicized, sometimes not italicized. You should keep the same single line. as a rule, the name of the genus is not italicized, only the name of the entire species (but which is missing here)

Line: 404: Related to: <...we utilized existing model species (Arabidopsis and rice) for the identification... >

Please refer strictly to the species and not the genus (as you did throughout the manuscript), especially in the discussions and when you mention it for the first time in a chapter. Also mention which rice variety you are discussing. This is too general and does not inspire real confidence.

Line 453: Please replace Triticum Urartu with Triticum Urartu

Line 543: Related to: <...we utilized existing model species (Arabidopsis and rice) for the identification... > 12 Oryza sativa (rice) and 10 Arabidopsis experimentally validated members... >

Give the species with the scientific name for Arabidopsis, as you did for Oryza sativa. Also, if you said there are 13 members, you should give details of them, who are they? I would omit the link and the referring site and I would describe it verbatim.

Line 620: Mention somewhere in Material and Method the Laboratory/s where you did these analyses. Place the activity in time and space so that it is credible.

Lines 639-631: <Related to: This study provides a basis 630 for subsequent functional studies of the GH13 family in wheat. >

Please develop what functional studies it is about. This is just a general statement.

Lines 644-645: I think you should say thank you by mentioning the person's name and that's it. What PhD thesis are you talking about? Is this study the subject of a doctoral thesis? Then specify this. Here something is not very clear.

Kind regards,

R

Author Response

Thank you for the reivew! Please see the attachment.

Reviewer 2 Report

Comments and Suggestions for Authors

This paper describes glycoside hydrolases from GH13 family, which is crucial for catalyzing α-glucoside linkages, pivotal in plant growth, development, and stress responses. Therefore, the topic of this manuscript (Ms) is relevant for IJMS. However, I have several critical remarks to the Ms. Ms needs Major Revisions.

Major:

1) Authors should increase the font size in Figure 1, 2, 3, 4, and 8. It is very difficult to understand those figures in its present form.

2) Authors should improve all used legends for figures and tables, e.g.

a) explain all used abbreviations;

b) in table 1: specify the GeneBank from which the gene sequences (Gene ID) are taken (www);

c) in figure 7: “The 10 stage are ‘germination’, ‘seedling growth’, ‘tillering’…” correct to “The 10 stage are ‘1, germination’, ‘2, seedling growth’, ‘3, tillering’…”. Also, add a numeric designation to the figure 7 itself.

3) Is it possibly to include the statistical treatment in Fig. 7 and 8?

4) Authors should explain in more detail how the data is obtained in Figure 7? The numbers in the GeneBank from where the transcription data was taken.

Minor:

5) Line 55: “mainly hydrolyze α-1,4 glucosidic bonds and α-1,6 glucosidic bonds in starch” correct to “mainly hydrolyze α-1,4 glucosidic and α-1,6 glucosidic bonds in starch”.

6) Line 111: “strong gluten and weak gluten varieties” correct to “strong and weak gluten varieties”.

Comments on the Quality of English Language

English:

This manuscript (Ms) contains misprints, mistakes in English grammar and in the writing style. I recommend that the authors should use some help of a native English speaker or send the Ms to an English Editing Service that proofreads scientific writing.

Author Response

(The authors gave the same response as above.)

Reviewer 3 Report

Comments and Suggestions for Authors

Dear Authors, please find attached Ms file for specific comments and queries.

Round 2

Reviewer 2 Report

Comments and Suggestions for Authors

Accept

Comments on the Quality of English Language

Minor editing of English language required